# Intellectual disability literacy and its connection to stigma: A multinational comparison study in three European countries

Elisabeth L. Zeilinger[1]*, Katharina A. M. Stiehl[1¤], Holly Bagnall[2], Katrina Scior[3]

1 Faculty of Psychology, University of Vienna, Vienna, Austria, 2 Department of Psychology, King's College London, London, United Kingdom, 3 Clinical, Educational and Health Psychology, University College London, London, United Kingdom

☯ These authors contributed equally to this work.
¤ D.O.T. Research Group for Mental Health of Children and Adolescents, Ludwig Boltzmann Society at Karl Landsteiner University of Health Sciences, Krems on the Danube, Austria
* elisabeth.zeilinger@univie.ac.at

**Data Availability Statement:** All data files are available from figshare (The respective DOI is: https://doi.org/10.6084/m9.figshare.12972884).

## Abstract

Article 8 of the UN Convention on the Rights of People with Disabilities requires governments around the globe to raise awareness on issues of disability in their societies, combatting stereotypes, prejudices and harmful practices. Little comparative data is available on lay people's social representations of intellectual disability and associated stigma, which could inform actions in line with the convention. The present study compared intellectual disability literacy and stigma among adults in the general population in the UK, Austria and Germany (N = 1046), and examined the effects of providing an intellectual disability label on these outcomes. Although participants came from highly developed European countries and prior contact with people with intellectual disabilities was common, the findings pointed to some striking differences between countries. Participants in Austria and Germany were much less likely than those in the UK to identify symptoms of a possible intellectual disability in an unlabelled vignette. They were also much more hostile towards social contact, a difference that was maintained when participants were informed of the intellectual disability label. Labelling showed controversial effects on stigma, both in its effects on social distance and on beliefs about suitable causes and interventions. Overall, the social acceptance of people with intellectual disabilities appears to be much less advanced in Austria and Germany than in the UK, and awareness of intellectual disability much lower, indicating a need for action at societal level.

## Introduction

People with intellectual disabilities (ID) are prone to being stigmatized and marginalized [1–3]. This has a direct negative effect on their quality of life and possibility for participation, contradicting the Convention on the Rights of Persons with Disabilities by the United Nations

**Funding:** Open access funding provided by University of Vienna.

**Competing interests:** The authors have declared that no competing interests exist.

(UN-CRPD) [4]. Stigma towards people with ID is considered to be understudied [5], thus, to tackle prejudice and stigma, it's important to learn more about the nature of stigma and its presentations towards people with ID.

Stigma is a very broad concept with a variety of definitions describing different presentations of stigma (such as cognitive, emotional, behavioural), as well as its consequences (such as stereotyping and prejudice) [6, 7]. Link and Phelan [8] define stigma as present, if: (a) human differences are noticed and labelled; (b) undesirable characteristics generate a stereotype connected to the labelled person in salient cultural beliefs, (c) people connected to the label are considered outsiders and (d) the consequence of the label is inequity and loss of status. Self-stigma directly affects people with a label, by internalizing other peoples' negative reactions, treatment and attitudes. A recent meta-analysis showed that most people with ID associate negative feelings, such as shame, powerlessness, and frustration with their ascribed label [1].

ID is also known as *Intellectual Developmental Disorder* in DSM-5 [9] or *Disorders of Intellectual Development* in ICD-11 [10]. It is characterized by limitations in intellectual functioning (IQ < 70) and in adaptive behaviour originating in the developmental phase of an individual [11]. Prevalence of ID in high income countries is estimated to be about 0.9% [12]. The diagnostic criteria of ID describe the impairment of the respective person. In the medical model of disability, the disability is understood to be the result of the impairment. In contrast, in the social model of disability, a person with an impairment does not have to be disabled. The disability is caused by a social organization, that excludes people with impairments from participating in community life. Central aspects of exclusion are stigma and negative attitudes towards persons with a disability [13, 14].

Societal views of people with ID and policies have changed substantially over the last hundred years. In the 1920s and 1930s the Eugenics movement claimed that people with ID were a danger to the moral and economical wealth of a nation and should be eliminated from the human gene pool. Although the Eugenics movement lost influence over time, as its early scientific assumptions were disproved by advanced research and concerns about the negative consequences of institutionalisation grew [15], the segregation of people with ID in large institutions continues to exist across Europe and in many other countries [16].

The right of persons with disabilities to full and equal enjoyment of all human rights and fundamental freedoms, as well as the governments duty to promote and protect these were only formally recognised very recently in the Convention on the Rights of Persons with Disabilities by the United Nations (UN-CRPD) [4]. These rights are mirrored in the social model of disability, which advocates for social and political change to create an environment with less barriers and increased understanding and acceptance of the needs of people with ID. In contrast, the medical model places the responsibility to fit into the already existing societal framework on the person with ID. In order to do so, people with ID require treatment to be "fixed and normalized" as far as possible [14].

An important route to ensuring equality for people with ID, and adhere to the social and human rights models of disability is to reduce prejudice and discrimination at societal level. Identifying to what extent lay perceptions and stigma differ across countries, cultural and religious communities can render useful evidence for public education efforts. The current paper focuses on the UK, Austria and Germany, countries which show similarities yet also marked differences in their history and in societal and policy responses to this population.

In the UK, deinstitutionalization began in the 1970s and the last large institutions for people with ID were closed in the 1990s. Current UK policies aim to maximise social inclusion, independence and empowerment [17, 18], and self-advocacy has become a powerful voice in the demand for change. The disability rights movement has had a vital role in challenging discrimination and advocating for legislation and policies that promote the equal rights and inclusion

of people with ID [19]. Nonetheless of adults with ID receiving support from local authorities, only 6% are in some form of employment [20]. Under the Nazi regime in Austria and Germany, alongside enforced sterilization on an unprecedented scale, people with ID were used for human experimentation or exterminated as one of the many groups deemed 'undesirable´ [21]. In order to protect the privacy of persons with ID after this horrible time, there are no official, publicly available registers and few national data concerning deinstitutionalization and service provision for this population [22]. The deinstitutionalization process started in the 1990s, approximately 20 years later than in the UK [23]. No national strategies were developed to successfully relocate persons with ID from large psychiatric hospitals to smaller residential facilities [16].

Nowadays, efforts are made to provide adequate support and housing for persons with ID in Austria and Germany [23]. Nevertheless, some facilities with more than 50 residents still exist, contradicting the principles of normalization and integration. Few people with ID are conventionally employed. Instead they mostly 'work' in day centres and sheltered workshops. Yet, since persons working in these centres are not included within employment statistics, hardly any information about the number of unemployed persons with ID and indicators of equality in the workforce is available.

After World War 2, attitudes towards persons with ID were still tainted by the ideas of the Nazi regime, which based people's relative worth on their ability to contribute to society [24]. As the question of compensation for forced sterilization arose, the claims of persons with ID, different to those of Jews, Sinti and Roma, were denied. Their sterilization was judged to have not been due to racial but eugenic reasons and therefore legally valid. Doctors who had sterilized people with ID for eugenic reasons were not prosecuted as their actions were within the law and not considered a crime against humanity [25].

As Austria and Germany share a similar recent history in which persons with ID were persecuted, this paper compares a mixed Austrian and German sample with a UK one. While the evidence would appear to suggest that lay people's attitudes towards this population have become more positive [26], our understanding of societal responses is limited by a lack of longitudinal data and few cross-cultural comparisons [27, 28].

A better understanding of public knowledge and beliefs about ID can help us to understand not only where but ideally also how attempts to reduce prejudice and discrimination against people with ID should be targeted. The present study set out to examine four key questions: (a) Do members of the general public in three European countries (the UK, Austria, and Germany) differ in their awareness of symptoms of ID, and what demographic characteristics influence this awareness? (b) Do persons differ in their beliefs about causes of ID and suitable interventions for an individual presenting with ID symptoms? (c) Do persons differ in their desire for social distance? (d) What are the effects of providing a diagnostic label on causal and interventional beliefs and on the desire for social distance?

## Method

### Participants

A total of 1046 participants participated in cross-sectional surveys conducted in London (UK sample, n = 526) and Vienna (Austria/Germany sample n = 520). In order to participate, individuals had to be aged 17 or above, and be a national or long-term resident of the respective country. The Austrian/German sample consisted of two thirds Austrians (n = 342) and one third Germans (n = 178). No significant differences between these two groups were observed at the 5% alpha level on any of the dependent variables, hence they were combined for all further analyses.

**Table 1. Sociodemographic characteristics and results of χ2-tests for differences between Austria/Germany and UK.**

| | Total sample (n = 1046) | | | | | | Sample with unlabelled vignette (n = 523) | | | | | |
|---|---|---|---|---|---|---|---|---|---|---|---|---|
| | UK (n = 526) | | Austria/Germany (n = 520) | | $\chi^2$ (1) | p | UK (n = 261) | | Austria/Germany (n = 262) | | $\chi^2$ (1) | p |
| | n | % | n | % | | | n | % | n | % | | |
| **Gender** | | | | | | | | | | | | |
| Female | 287 | 54.8 | 280 | 54.1 | 0.05 | .852 | 139 | 53.3 | 139 | 53.3 | -[a] | -[a] |
| Male | 237 | 45.2 | 238 | 45.9 | | | 122 | 46.7 | 122 | 46.7 | | |
| **Education** | | | | | | | | | | | | |
| Graduates | 198 | 37.7 | 117 | 24 | 22.28 | < .001*** | 69 | 26.5 | 49 | 20.1 | 2.93 | .087 |
| Non-graduates | 327 | 62.3 | 371 | 76 | | | 191 | 73.5 | 195 | 79.9 | | |
| **Religion** | | | | | | | | | | | | |
| Religious | 219 | 42 | 358 | 69.3 | 78.35 | < .001*** | 100 | 38.9 | 185 | 70.9 | 53.48 | < .001*** |
| Non-religious | 303 | 58 | 159 | 30.7 | | | 157 | 61.1 | 76 | 29.1 | | |
| **Prior contact with ID** | | | | | | | | | | | | |
| Yes | 209 | 44.2 | 328 | 63.1 | 35.59 | < .001*** | 111 | 53.4 | 159 | 60.7 | 2.54 | .133 |
| No | 264 | 55.8 | 192 | 36.9 | | | 97 | 46.6 | 103 | 39.3 | | |

*Note.*

[a] not computable, as the distribution in both samples was exactly the same

*p<0.05

** p<0.01

*** p<0.001.

UK and Austrian/German participants were comparable in respect to age and gender. The UK sample contained more graduates, and fewer persons who self-identified as religious. Prior contact to persons with ID was reported more frequently by Austrian/German participants. However, among those who reported prior contact, UK respondents reported a significantly closer relationship with the respective person with ID. Demographic characteristics and test statistics of differences between the two samples are depicted in Tables 1 and 2.

The subsample of respondents presented with the unlabelled vignette only showed differences in religiosity and in the closeness of prior contact to a person with ID. UK respondent were less religious but rated their relationships to people with ID as closer.

## Measures

**Intellectual disability literacy scale.** For the assessment of ID literacy and stigma, the *Intellectual Disability Literacy Scale* (IDLS) [29] was used. This instrument has sound psychometric properties [29] and comprises three parts, namely recognition (1 item), causal and intervention beliefs (44 items), and social distance (4 items). In the first part of the IDLS, participants were presented with a diagnostically labelled or unlabelled vignette of a man in his

**Table 2. Sociodemographic characteristics and results of t-tests for differences between Austria/Germany and UK.**

| | Total sample (n = 1046) | | | | | | | Sample with unlabelled vignette (n = 523) | | | | | | |
|---|---|---|---|---|---|---|---|---|---|---|---|---|---|---|
| | UK (n = 526) | | Austria/Germany (n = 520) | | t | df | p | UK (n = 261) | | Austria/Germany (n = 262) | | t | df | p |
| | M | SD | M | SD | | | | M | SD | M | SD | | | |
| **Age** | 31.34 | 11.41 | 31.69 | 13.3 | 0.45 | 1013 | .649 | 30.07 | 11.98 | 31.58 | 13.57 | 1.35 | 514 | .177 |
| **Closeness of contact with persons with ID** | 4.34 | 2.62 | 3.66 | 2.46 | 3.04 | 535 | .002** | 4.35 | 2.49 | 3.68 | 2.52 | 2.17 | 268 | .031* |

early 20s, who met ICD-10 diagnostic criteria for a (mild) ID [30], as described in Scior and Furnham [29]. Of the UK sample 265 (50.38%) participants received the labelled version, and 258 (49.62%) of the Austrian/German sample. Both vignette versions were identical other than inclusion/omission of a diagnostic label, consisting of 'learning disability (mental handicap)' in the UK and 'intellektuelle Behinderung (geistige Behinderung)' in Austria and Germany. Following presentation of the vignette, for the unlabelled version, participants were asked "what would you say is going on with X?'. If multiple responses were given, the response closest to the correct diagnosis was registered.

The second part of the IDLS assesses causal and interventional beliefs. Respondents rated their agreement with 22 potential causes of the behaviours depicted in the vignette and 22 sources of help, using a 7-point Likert scale (1 = strongly disagree to 7 = strongly agree). The causal items relate to four subscales: biomedical (5 items), environmental (7 items), adversity (5 items) and supernatural (5 items). The intervention items relate to three subscales: expert help (6 items), lifestyle changes (11 items) and religious/spiritual sources of help (5 items). Higher scores indicate increased agreement.

The third part of the IDLS comprises items related to social distance. This scale was used to assess stigma. Although social distance does not capture all aspects of stigma, it is regarded a measure of external stigma [8, 29]. Participants rated their views on social contact with someone like the person described in the vignette by responding to four statements about social contact in situations of increasing intimacy (live next door, spend an evening socialising, make friends, have the person marry into one's family), using a 7-point fully anchored Likert scale (1 = disagree strongly to 7 = agree strongly). A social distance score was calculated as a mean of reversed responses, with higher scores indicating a stronger desire for social distance.

**Contact and sociodemographic variables.**   Prior contact to persons with ID was assessed via two items: whether the respondent ever had contact to a person with ID (yes/no), and where affirmed, how close this contact was (rated on a 9 point scale where 1 = *not at all close*, 5 = *somewhat close* and 9 = *extremely close*). Additionally, we assessed age, gender, educational attainment, and whether the respondents viewed themselves as religious.

## Procedure

For the survey in Austria and Germany, the materials were translated from the original English version into German-language using a parallel-blind technique followed by a back translation method [31]. A professional translator helped finalise the translation and discrepancies were resolved through discussion.

For data collection, cross-sectional surveys were conducted in London (UK sample) between July 2009 and January 2010, and in Vienna (Austria/Germany sample) between October 2011 and January 2012. No specific inclusion or exclusion criteria were applied for participant recruitment. Potential participants were approached either in person or via email circulation lists with a link to an anonymous web survey. They were asked to complete a brief survey on their views of personal difficulties in others and invited to share the invitation with others. Participants consented to participation after reading the study information and before proceeding to the survey. Those approached in person returned their paper version of the survey anonymously and separate from the consent form. Overall, 84% of respondents completed the web version, the remainder the paper version. It was not possible to calculate a response rate, given that the total number who received an email invitation could not be tracked. Participants were randomly assigned the labelled or unlabelled vignette. The UK dataset was initially collected for a validation study of the IDLS, and comprised 1,376 participants. To minimize the influence of demographic characteristics on the main outcomes, we used a stratified

random sample selected from this larger data set, blind to variables other than participants' key demographics As we used a convenience sampling strategy, the samples cannot be regarded representative of the larger population. Ethical approval for the study was granted by the University College London Research Ethics Committee, Project ID: 0960/001.

## Data analysis

To examine differences between the UK and Austrian/German sample, $\chi^2$ tests, t-tests and ANOVAs were computed. A multiple regression analysis was performed to determine influencing factors on ID literacy and social distance. We used an alpha level of 5% to determine statistically significant results. Effect sizes are interpreted following the recommendations of Cohen [32].

Exploration of the data revealed that both the supernatural causal beliefs subscale and religious interventions subscale showed significant positive skewness, i.e. overall participants tended to disagree. Consequently, scores on both were log-transformed. All statistical analyses were carried out on the log-transformed data, but the original means and standard deviations of these subscales are reported to ease interpretation.

## Results

### Awareness of symptoms of intellectual disability

To compare awareness of symptoms of mild ID in the different localities, only the subsample that was presented the unlabelled vignette was examined. Of the 261 UK participants, 38.3% recognised a possible ID compared to only 12.4% of the 262 Austrian/German respondents ($\chi^2$ (1) = 45.93, $p < .001$).

### Causal beliefs

The internal reliability of all causal belief subscales was good for both samples, with Cronbach's α between .71 and .89, see Table 3.

To compare UK and Austria/Germany subsamples that received the labelled vignette (i.e. who were in no doubt about the diagnosis; $n = 523$) we computed independent sample t-tests. To assess the influence of country and diagnostic labelling on causal beliefs, we computed independent factorial ANOVAs on the whole sample ($N = 1046$).

Looking at the labelled subsample only, Austrian and German participants were more likely to endorse biomedical, adversity and environmental causes, all significant at $p < .01$ once Bonferroni corrected. No difference between the two samples was observed for supernatural causes.

**Table 3. Cronbach's alphas for all scales and subscales.**

|  | Whole Sample N = 1046 | UK n = 526 | Austria/Germany n = 520 |
|---|---|---|---|
| **Causal Beliefs** |  |  |  |
| Biomedical | .88 | .86 | .89 |
| Adversity | .80 | .81 | .78 |
| Environmental | .85 | .86 | .84 |
| Supernatural | .71 | .73 | .70 |
| **Intervention Beliefs** |  |  |  |
| Expert help | .76 | .76 | .76 |
| Lifestyle | .82 | .86 | .81 |
| Religious/Spiritual | .76 | .79 | .81 |
| **Social Distance** | .89 | .88 | .85 |

**Table 4. Causal beliefs, intervention beliefs and social distance by country and condition: Means (standard deviations).**

| Variable | UK | | | Austria/Germany | | |
|---|---|---|---|---|---|---|
| | All N = 526 | Unlabelled N = 261 | Labelled N = 265 | All N = 520 | Unlabelled N = 262 | Labelled N = 258 |
| **Causal Beliefs** | | | | | | |
| Biomedical | 4.14 (1.37) | 3.62 (1.37) | 4.65 (1.17) | 4.33 (1.71) | 3.50 (1.54) | 5.18 (1.43) |
| Adversity | 3.16 (1.27) | 3.47 (1.27) | 2.86 (1.20) | 3.77 (1.35) | 3.91 (1.34) | 3.63 (1.35) |
| Environmental | 2.93 (1.26) | 3.46 (1.17) | 2.40 (1.13) | 3.44 (1.40) | 4.00 (1.20) | 2.89 (1.39) |
| Religious/Spiritual | 1.57 (0.82) | 1.76 (0.89) | 1.40 (0.69) | 1.60 (0.85) | 1.70 (0.91) | 1.52 (0.77) |
| **Intervention Beliefs** | | | | | | |
| Expert help | 4.26 (1.20) | 4.26 (1.19) | 4.26 (1.21) | 4.61 (1.30) | 4.40 (1.33) | 4.82 (1.23) |
| Lifestyle | 3.83 (1.08) | 4.17 (1.13) | 3.50 (.91) | 3.89 (1.04) | 4.10 (1.03) | 3.69 (1.03) |
| Religion | 2.22 (1.12) | 2.13 (1.26) | 2.31 (0.96) | 1.86 (1.06) | 2.00 (1.14) | 1.72 (0.96) |
| **Social Distance** | | | | | | |
| | 3.57 (1.46) | 3.82 (1.47) | 3.32 (1.41) | 5.09 (1.23) | 5.30 (1.50) | 4.89 (1.28) |

There was a small main effect of country on biomedical causal beliefs, $F(1, 1042) = 5.25$, $p = 0.22$, $\eta^2 = .01$, and a large effect of labelling, $F(1, 1042) = 255.00$, $p < 0.001$, $\eta^2 = .20$. There was also a small interaction between country of origin and labelling on biomedical causal beliefs, $F(1, 1042) = 15.31$, $p < 0.001$, $\eta^2 = .01$. Austrian and German participants were more likely to endorse biomedical causes than UK participants, see Table 4. Those presented with the labelled vignette ($M = 4.92$, $SD = 1.33$) were more likely to agree with biomedical causes than those presented with the unlabelled vignette ($M = 3.55$, $SD = 1.46$). The effect of labelling was more pronounced for the Austrian and German sample.

For adversity causal beliefs there was a small main effect of country, $F(1, 1042) = 57.35$, $p < 0.001$, $\eta^2 = .05$, and labelling, $F(1, 1042) = 30.74$, $p < 0.001$, $\eta^2 = .03$. There was also a small interaction between country and labelling on adversity causal belief scores, $F(1, 1042) = 4.21$, $p = 0.04$, $\eta^2 = .004$. Austrian and German participants were more likely to endorse adversity causes than UK respondents. Those given the unlabelled vignette were more likely to endorse adversity causes ($M = 3.69$, $SD = 1.32$) than the labelled group ($M = 3.24$, $SD = 1.33$). Labelling had a somewhat larger effect in reducing endorsement of adversity causes among UK participants.

For environmental causal beliefs there were small main effects of country, $F(1, 1042) = 43.36$, $p < 0.001$, $\eta^2 = .04$, and a large effect of labelling, $F(1, 1042) = 199.80$, $p < 0.001$, $\eta^2 = .16$. The interaction between country and labelling was not significant, $F(1, 1042) = 0.02$, $p = .89$. Austrians and Germans were more likely to endorse environmental causes than UK respondents, as were those presented with the unlabelled vignette ($M = 3.72$, $SD = 1.21$) compared to the labelled group ($M = 2.64$, $SD = 1.28$).

Finally, for supernatural causal beliefs there was no main effect of country, $F(1, 1042) = 0.38$, $p = .54$, but a small main effect of labelling, $F(1, 1042) = 32.99$, $p < 0.001$, $\eta^2 = .03$, and a small interaction between country and labelling, $F(1, 1042) = 6.78$, $p = .01$, $\eta^2 = .01$. Those given the unlabelled vignette were more likely to agree with supernatural causes ($M = 1.72$, $SD = 0.90$) than the labelled group ($M = 1.45$, $SD = 0.73$). Labelling had a larger suppressing effect on endorsement of supernatural causes among UK participants.

### Intervention beliefs

The internal reliability of all intervention subscales was good for both samples, with Cronbach's α between .76 and .86, see Table 3. Again, we compared responses between the UK and Austria/Germany only for those subsamples that received the labelled vignette using

independent sample t-tests. Subsequently, to assess the influence of country and diagnostic labelling on intervention beliefs independent factorial ANOVAs were conducted for the entire sample.

Looking at the labelled subsamples only, Austrian and German participants were more likely to agree that expert help might be beneficial and less likely to view religious interventions as helpful ($p < .01$, Bonferroni corrected).

There was a small main effect of country on endorsement of expert led interventions, $F(1, 1042) = 20.69$, $p < 0.001$, $\eta^2 = .02$, and of labelling, $F(1, 1042) = 7.65$, $p = .006$, $\eta^2 = .01$. There was also a small interaction effect between country and labelling, $F(1, 1042) = 8.04$, $p = 0.005$, $\eta^2 = .01$. Austrians and Germans were more likely to view expert interventions as helpful than were UK respondents, as were those given the labelled vignette ($M = 4.54$, $SD = 1.25$) compared to the unlabelled group ($M = 4.33$, $SD = 1.26$). Labelling strongly increased endorsement of expert help among the Austrian/German sample but had no effect in the UK.

For lifestyle changes there was no significant main effect of country, $F(1, 1042) = 0.84$, $p = .36$, but a modest main effect of labelling, $F(1, 1042) = 71.35$, $p < 0.001$, $\eta^2 = .06$. There was also small interaction between country and labelling, $F(1, 1042) = 4.36$, $p = .04$, $\eta^2 = .01$. Those given the unlabelled vignette were more likely to agree that lifestyle changes might be helpful ($M = 4.13$, $SD = 1.08$) than the labelled group ($M = 3.60$, $SD = 0.97$). Labelling had a stronger suppressing effect on endorsement of lifestyle changes among the UK sample.

For religious/spiritual interventions there was a small main effect of country, $F(1, 1042) = 38.79$, $p < 0.001$, $\eta^2 = .04$, but no significant main effect of labelling, $F(1, 1042) = 0.004$, $p = .95$. There was a small interaction effect between country and labelling, $F(1, 1042) = 20.03$, $p < 0.001$, $\eta^2 = .02$. Austrians and Germans were a little less likely to endorse such interventions than UK respondents. Whilst there was no difference *per se* between those presented with the labelled vignette ($M = 2.02$, $SD = 1.00$) and the unlabelled vignette ($M = 2.07$, $SD = 1.20$), labelling had the opposite effect on beliefs about the suitability of such interventions in both countries; among UK participants agreement with religious/spiritual interventions was stronger among those presented with the labelled vignette, yet in Austria and Germany the opposite was observed. While this result seems surprising, it may be explained by the finding that contact, which differed between the two countries, predicted agreement with religious interventions.

## Social distance

The internal reliability of the social distance scale was high for both samples, with Cronbach's α of .85 or above, see Table 3. Looking at the labelled subsample only, Austrian and German participants were more likely to keep social distance from someone with an ID ($p < .01$, Bonferroni corrected).

Independent factorial ANOVAs revealed a significant large main effect on social distance scores of country, $F(1, 1042) = 338.35$, $p < 0.001$, $\eta^2 = .25$, and a small effect of labelling, $F(1, 1042) = 29.90$, $p < 0.001$, $\eta^2 = .03$. There was no significant interaction between country and labelling on social distance, $F(1, 1042) = 0.34$, $p = .56$. Austrians and Germans scored higher on social distance than UK participants. Those given the unlabelled vignette showed a higher desire for social distance ($M = 4.56$, $SD = 1.51$) than the labelled group ($M = 4.10$, $SD = 1.56$).

## Influence of participant characteristics on intellectual disability literacy and social distance

To determine which aspects influence ID literacy and social distance, and to examine whether some of the findings observed could be due to characteristics of the samples, regression

**Table 5. Odds ratios (95% confidence intervals) from logistic regression analyses: Effects of socio-demographic characteristics and contact on the likelihood of identifying intellectual disability.**

| Variable | B | SE B | OR |
|----------|-----|------|-----|
| Constant | 0.47 | 0.37 | N/A |
| Country | 1.57 | 0.25 | 0.21*** (0.13–0.34) |
| Gender | 0.73 | 0.24 | 0.48** (0.30–0.78) |
| Contact | 0.56 | 0.25 | 1.76* (1.08–2.85) |
| Age | 0.00 | 0.01 | 1.00 (0.98–1.02) |
| Education | 0.03 | 0.30 | 1.03 (0.58–1.84) |

*Note.* pseudo-$R^2$ = 17.8% (Nagelkerke). Model $\chi^2(5)$ = 56.88, $p<0.001$.

Country: 0 = UK; 1 = Austria/Germany; Gender: 0 = female, 1 = male; Contact: 0 = no prior contact, 1 = prior contact; Education: 0 = to age 18 or less, 1 = graduate.

*$p<0.05$

** $p<0.01$

*** $p<0.001$.

analyses were computed. Predictors of recognition of a possible ID in the unlabelled vignette were identified using logistic regression, see Table 5. This showed that country was the strongest predictor, with gender and contact also playing a role. UK residents, women and those with prior contact with someone with an ID were more likely to identify ID.

The results regarding predictors of social distance are shown in Table 6. Social distance was predicted by country, prior contact and recognition. Austrian and German participants, those without prior contact and those who failed to detect a possible ID were more likely to stigmatise the individual depicted in the vignette.

## Discussion

This study set out to compare lay people's awareness, beliefs and stigma regarding ID in three highly developed Western European countries, the UK, Austria, and Germany. Notwithstanding differences in policy and service provision in the three countries, despite good intentions of legislators and policy makers, society still seems to struggle with welcoming people with ID

**Table 6. Predictors of social distance: Results of multiple regression analyses.**

| | B | SE B | β |
|---|-----|------|-----|
| **Social Distance** | | | |
| Constant | 4.05 | .17 | |
| Recognition | -0.35 | .15 | -.10* |
| Country | 1.42 | .13 | .47*** |
| Gender | 0.2 | .12 | .07 |
| Contact | -0.33 | .12 | -.11** |
| Religion | -0.01 | .03 | -.01 |

*Note.* Social Distance $R^2$ = 27% ($p<0.001$).

Recognition: 0 = no, 1 = yes; Country: 0 = UK; 1 = Austria; Gender: 0 = female, 1 = male; Contact: 0 = no prior contact, 1 = prior contact.

*$p<0.05$

** $p<0.01$

*** $p<0.001$.

as equals in all three countries. The aims of the social and human rights models of disability have not yet been accomplished. Even in the UK, where stigma was lower, most lay people were ambivalent about social interactions with people with ID.

We found some striking differences between the UK and Austria/Germany. Awareness of symptoms of a possible mild ID was considerably lower in Austria/Germany than in the UK (12% versus 38%). ID stigma was a lot higher among the Austrian/German sample. Considering that 63% of participants in Austria/Germany and 44% in the UK reported at least some prior contact with people with ID, these findings contradict previous evidence showing contact to have a positive effect on attitudes [33–36]. Other studies with culturally diverse samples in the UK, using the same methodology, found much higher recognition rates, as well as a much lower desire for social distance compared to the present Austrian/German sample [36]. Thus, stigma could be considered a major barrier to the greater social inclusion of people with ID, particularly in Austria and Germany.

Some factors explaining the current findings may include the shared history during the Nazi regime in Austria and Germany, later deinstitutionalization and lower attention to policy and service delivery pertaining to people with ID, including missing public awareness campaigns related to persons with ID and their rights. As the disability rights movement has been powerful in Britain for decades, it may not be surprising that the UK has an advantage over other European countries in terms of respecting the rights of people with disabilities and taking adequate actions in terms of awareness raising. The question whether stigma towards marginalised groups may generally, not just for people with ID, be lower in the UK than in Austria and Germany is impossible to answer as there are no respective comparative studies to draw on. Of note, while the majority of Austrian and German participants in both the labelled and unlabelled conditions was opposed to social contact, UK participants were typically ambivalent about social contact, rather than positively welcoming it.

Although social contact has been shown to be most effective to reduce stigma and negative attitudes towards persons with ID [34, 35], efforts to reduce ID stigma are sparse and need to be more prominent [3]. This is especially alarming, when taking the negative consequences of stigma and the implications for people with ID into account [37]. The ID field could learn from interventions against mental-health stigma, which are far more frequent and well evaluated [38].

With regards to beliefs about possible causes, Austrians and Germans were more likely than UK participants to endorse biomedical, adversity and environmental causes than UK participants. They were more likely to view expert help as potentially helpful and less likely to view religious interventions, which received low endorsement in both countries, as suitable. Endorsement of expert help and biomedical causes might indicate a higher backing of the medical model of disability in Austria and Germany compared to the UK. In the medical model experts and professionals are considered to be the agents for remedy and disability is generally viewed as negative. Endorsement of the medical model might therefore lead to a higher wish for social distance and further stigma.

Notwithstanding concerns voiced about the negative effects of labelling, providing lay people with an ID label in the current study was associated with reduced stugma. Similar to a a recent study of people with ID with/without comorbid schizophrenia, those informed of the diagnostic label were more positive towards social contact than those who provided their own, potentially more stigmatising explanations for the person's behaviour [5]. Labelling was also associated with increased endorsement of biomedical causes and reduced endorsement of environmental causes. Of note, biomedical attributions for ID and rejection of environmental causes have been shown to be associated with reduced stigma, possibly because the person is less likely to be held responsible for their difficulties if these are attributed to a disability and

biomedical causes beyond the person's control [39]. Conversely, endorsement of biomedical causes could be linked to an underlying endorsement of the medical model of disability. One further reason for the labelled vignette eliciting less desire for social distance could be that the general public tends to view people with ID with sympathy, or much more problematically pity, rather than fear [39, 40]. This merits further investigation, as contact with a person with ID motivated by pity would hardly result in a positive contact experience for the person with ID and reinforce the devaluation of this population.

Labelling was also associated with increased endorsement of expert help and reduced endorsement of lifestyle changes. These findings may be viewed as an acknowledgement of the need for skilled support. They could also be linked to endorsement of the medical model of disability, where persons with ID are seen as in need of cure, or need to get fixed, rather than in need of empowerment and support for self-determination.

Stereotyping can be the basis for stigma; however, it was recently shown, that stereotypes of persons with ID are not related to high levels of explicit discrimination [2]. Furthermore, the fact that the case vignette without an ID label elicited greater agreement with lifestyle solutions and a reduced belief in the need for expert helpt (in Austria and Germany only) may also point to diagnostic overshadowing [41, 42].

In the present study, effects of labelling depended on the country for three of the four causal beliefs subscales and all three interventions beliefs subscales. While the interaction effects observed were small, they suggest that the effects of applying diagnostic labels should not be assumed to be universal as they may well be country or culture specific. In the absence of research that has explored why this may be, one might conjecture that the different effects of labelling relate to different interpretations of what it means to have an ID, and thus perhaps reflect broader attitudinal and policy differences between countries.

## Limitations

Several limitations of the current study should be noted. First, arguably the results for the samples that received the unlabelled vignette should be viewed with caution as most participants failed to recognise a possible ID and hence their social distance and belief ratings referred to other presumed causes of the difficulties presented. We attempted to address this issue by presenting half of the sample with an identical diagnostically labelled version of the vignette and using a factorial design in the analysis. A further important limitation to note is that our findings denote associations and not cause and effect relationships, which would require experimental studies that are unfortunately scarce to date in the ID field.

In both locations a convenience sampling approach was employed. While we attempted to ensure that participants in both samples, UK and Austria/Germany, were matched on socio-demographic characteristics to allow for a meaningful comparison, caution should be exercised in generalising the findings to the general public, as the samples cannot be regarded representative for a larger population. The rates of prior contact with individuals with ID reported in Austria and Germany appear untypically high and hint at a potential self-selection bias. In the UK sample graduates at 38% of the sample were overrepresented compared to the 23% found in the general UK population [43].

Aspects like ethnicity, being a professional working with persons with ID or having a relative with ID can influence ID literacy and stigma. However, these aspects were not assessed in this study, and could not be taken into account in our analysis. Future work should address these aspects in more detail.

Finally, we should note that thisstudy focused on explicit attitudes and beliefs, which are based on self-report. Although most previous studies that have investigated whether social

desirability affects self-reported attitudes to people with ID found no such bias [26], it cannot be guaranteed that the results are a genuine reflection of respondents' attitudes or their likelihood to behave in certain ways in real life situations. There is some evidence that explicit attitudes differ from implicit attitudes [44], but more research needs to be done on studying implicit attitudes and on examining the relationship between self-reported attitudes and actual behaviour.

## Implications

Although the vignette intentionally presented someone with a mild ID, and thus with signs that may not be obvious to lay people, the fact that 62% of participants in the UK and 88% in Austria and Germany failed to detect a possible ID indicates that more needs to be done to raise public awareness of ID. Most people with ID experience mild deficits. The risk that many lay people may misattribute their difficulties to other potentially more stigmatising causes highlights the need for efforts in this regard. The suggestion of high levels of stigma associated with ID in Austria and Germany in particular indicates a need for work aimed at removing barriers to successful integration and the assurance of equal rights of people with ID, in line with obligations under the UN Convention on the Rights of Persons with Disabilities.

## Supporting information

**S1 File. Labelled vignettes of the Intellectual Disability Literacy Scale (IDLS).**
(PDF)

## Author Contributions

**Conceptualization:** Elisabeth L. Zeilinger, Katrina Scior.

**Formal analysis:** Elisabeth L. Zeilinger, Katharina A. M. Stiehl, Holly Bagnall, Katrina Scior.

**Investigation:** Elisabeth L. Zeilinger, Katharina A. M. Stiehl, Holly Bagnall, Katrina Scior.

**Methodology:** Elisabeth L. Zeilinger, Katrina Scior.

**Project administration:** Elisabeth L. Zeilinger, Katrina Scior.

**Supervision:** Elisabeth L. Zeilinger, Katrina Scior.

**Writing – original draft:** Elisabeth L. Zeilinger, Katharina A. M. Stiehl, Holly Bagnall, Katrina Scior.

**Writing – review & editing:** Elisabeth L. Zeilinger, Katharina A. M. Stiehl, Holly Bagnall, Katrina Scior.

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
