## [Decision Letter · Decision Letter 0]

26 Jun 2020

PONE-D-20-07449

Intellectual disability literacy and its connection to stigma: A multinational comparison study in three European countries

PLOS ONE

Dear Dr. Zeilinger,

Thank you for submitting your manuscript to PLOS ONE. After careful consideration, we feel that it has merit but does not fully meet PLOS ONE’s publication criteria as it currently stands. Therefore, we invite you to submit a revised version of the manuscript that addresses the points raised during the review process.

The reviewers ask some clarifications, and some more attention for  the disability studies literature and disability rights movement.

We look forward to receiving your revised manuscript.

Kind regards,

Therese van Amelsvoort

Academic Editor

PLOS ONE

Journal Requirements:

2. Please provide additional information about the participant recruitment method and the demographic details of your participants. Please provide such additional information as a) the recruitment date range (month and year), b) a description of any inclusion/exclusion criteria that were applied to participant recruitment, c) a table of relevant demographic details, d) a statement as to whether your sample can be considered representative of a larger population, e) a description of how participants were recruited, and f) descriptions of where participants were recruited and where the research took place.

"Open access funding provided by University of Vienna."

"The authors received no specific funding for this work."

Reviewers' comments:

Reviewer's Responses to Questions

**Comments to the Author**

1. Is the manuscript technically sound, and do the data support the conclusions?

Reviewer #1: No

Reviewer #2: Yes

2. Has the statistical analysis been performed appropriately and rigorously? 

Reviewer #1: Yes

Reviewer #2: Yes

3. Have the authors made all data underlying the findings in their manuscript fully available?

Reviewer #1: Yes

Reviewer #2: Yes

4. Is the manuscript presented in an intelligible fashion and written in standard English?

Reviewer #1: Yes

Reviewer #2: Yes

5. Review Comments to the Author

Reviewer #1: Thank you for the opportunity to review your manuscript. Teh authors broach an incredibly important topic: the stigmatization of people with ID. I think this paper could be substantially improved by:

1) incorporation of literature on the social model of disability (vs the medical model). The disability studies literature is centered on the issue of stigmatization and has presented an alternative view of disability (social model)--one in which the problem is not the body, but within society itself--and its ableist attitudes and structures. In other words, disability is socially constructed and reflects the biases of the "able-bodied" and cultural/socially influenced assumptions about normality. This literature is not mentioned or cited in the manuscript. In a sense, the voices of disability scholars are marginalized in this manuscript.

2) The medical model (which the disability rights community rejects) suggests that diagnostic labels and the focus on medical impairment are problematic. Yet, the manuscript suggests that diagnostic label (a medical model type understanding of disability) results in less stigma? This is perplexing, and perhaps reflects conflicting perceptions of disability and stigma. Interestingly, some in the disability community might argue that the notion of intellectual disability is a social construction--that has evolved over time and reflects shifting societal assumptions/understandings about "normality".

3) The authors do not explicitly define how stigma is measured-although it appears to be measured by the social distance variable. The operationalization of stigma must be explicit and justified.

4) At one point, the authors talk about the positive effects of labeling and its effect on social distancing. They mention that pity/sympathy might be an effect of the label, thereby reducing the wish for social distancing. In the world of disability studies, there is nothing worse than pity. The rallying cry of disability advocates in the disability rights movement was "piss on pity". I cannot see how social contact, as a result of pity, would be a positive.

5) Notion of requiring expert help as destigmatizing---is reflective of the medical model of disability in which disabled people are "sick" and need of a "cure". This in itself is stigmatizing as perceived by the disability rights community.

6) Not convinced that differences in attitudes by country is a result of Nazi past. UK and Germany have very distinct health care systems and welfare regimes. More importantly, the UK had a very visible, powerful disability rights movement--in fact British disabled advocates coined the social model of disability. They were leaders in the field. The movement essentially began in Britain and inspired US advocates to forge their own movement. This is not mentioned and likely plays a role in public attitudes. Failure to adequately address the disability rights movement--and its impact on attitudes-- is a major omission in this manuscript.

Thank you again for the opportunity to review the manuscript. I think the study could be improved by greater understanding and inclusion of the disability studies literature and disability rights movement.

Reviewer #2: This is an interesting study that compared intellectual disability literacy and stigma among adults in the general population in the UK, Austria and Germany, and examined the effects of providing an intellectual disability label on these outcomes. Stigma is still an important societal issue. The paper is clearly written and methodologically sound.

I have few comments / questions

Subject demographics:

How where subjects recruited exactly? Authors state that for the UK sample they used a stratified sample selected from a larger data set and that potential participants were approached either in person or via email circulation lists with a link to an anonymous web survey.

How was the original sample recruited and for what purpose? Is this a representative sample for larger society? Please provide more information.

In addition, were other demographic variables that could influence the outcome measures such as ethnicity or religion assessed? Did participants had relatives with ID of were mental health professionals? Such factors could have influences participants ID literacy and stigma.

The authors stat that it was not possible to calculate a response rate, given that the number of persons who received an email invitation could not be tracked. This seems a bit odd, given that they stratified a sample based on a previous data set (at least for UK sample).

Did the authors also explore differences in outcome measures for the Austrian and German sample? Although generally comparable, there are likely to be more subtle cultural differences or differences in government polices possibly affecting the outcome measures.

Can the authors provide the vignette and the causal items participants had to rate? Perhaps in a supplement. Did the vignette also include behavioral symptoms?

6. PLOS authors have the option to publish the peer review history of their article (what does this mean?). If published, this will include your full peer review and any attached files.

Reviewer #1: **Yes: **Kristin Berg

Reviewer #2: No

---

## [Author Response · Author response to Decision Letter 0]

17 Jul 2020

Ms. Ref. No.: PONE-D-20-07449

To the Editor — PLOS ONE

List of changes in the revised manuscript and comments in response to Editorial Letter dated June 26, 2020:

Journal Requirements:

ad 1.) Changes made in accordance with style requirements:

- We changed the file naming 

- Title page: We added initials to the e-mail address of the corresponding author (ELZ)

- We deleted the numbering of headings and subheadings and formatted the font size of headings according to the style requirements.

- We changed the current affiliation of one author, Katharina A.M. Stiehl.

2. Please provide additional information about the participant recruitment method and the demographic details of your participants. Please provide such additional information as a) the recruitment date range (month and year), b) a description of any inclusion/exclusion criteria that were applied to participant recruitment, c) a table of relevant demographic details, d) a statement as to whether your sample can be considered representative of a larger population, e) a description of how participants were recruited, and f) descriptions of where participants were recruited and where the research took place.

ad 2.) We added more information to the methods section, including (a) the precise recruitment date range, (b) exclusion/inclusion criteria, (c) two tables with demographic details, (d) a statement that the present sample cannot be regarded as representative of the larger population, (e) a detailed description of the convenience sampling procedure for recruiting participants, and (f) that the research was conducted in Vienna for the Austrian/German sample, and in London for the UK sample.

ad 3.) We added more information to the measures section describing the questionnaire. We added the information, that the questionnaire used in this study, the IDLS, is already published:

Scior K, Furnham AF. Development and validation of the intellectual disability literacy scale for assessment of knowledge, beliefs and attitudes to intellectual disability. Res Dev Disabil. 2011;32: 1530–1541

However, since the vignettes are the central part of the questionnaire, we uploaded them as supplement to the revised version of our manuscript in both, German and English.

"Open access funding provided by University of Vienna."

"The authors received no specific funding for this work."

ad 4.) We deleted the comment about funding in the acknowledgement section. The new funding statement should read "Open access funding provided by University of Vienna.".

ad 5.) Our data file is now available at OPEN ICPSR. The respective DOI is: https://doi.org/10.3886/E118161V1

We changed the statement via the submission system.

Reviewer #1: 

Thank you for the opportunity to review your manuscript. Teh authors broach an incredibly important topic: the stigmatization of people with ID. I think this paper could be substantially improved by:

1) incorporation of literature on the social model of disability (vs the medical model). The disability studies literature is centered on the issue of stigmatization and has presented an alternative view of disability (social model)--one in which the problem is not the body, but within society itself--and its ableist attitudes and structures. In other words, disability is socially constructed and reflects the biases of the "able-bodied" and cultural/socially influenced assumptions about normality. This literature is not mentioned or cited in the manuscript. In a sense, the voices of disability scholars are marginalized in this manuscript.

ad 1.) Thank you for this comment. We are deeply sorry for leaving the impression that we marginalize the voices of persons with disabilities! We included the discrepancies between the social and medical model of disability in the introduction and discussion section.

2) The medical model (which the disability rights community rejects) suggests that diagnostic labels and the focus on medical impairment are problematic. Yet, the manuscript suggests that diagnostic label (a medical model type understanding of disability) results in less stigma? This is perplexing, and perhaps reflects conflicting perceptions of disability and stigma. Interestingly, some in the disability community might argue that the notion of intellectual disability is a social construction--that has evolved over time and reflects shifting societal assumptions/understandings about "normality".

ad 2.) The data of our study indicated, that the intellectual disability label led to more positive attitudes toward social contact with the person described in the vignette (the person with intellectual disability). We offer various alternative explanations this finding. In the revised manuscript, we included the medical model in the interpretation of this finding.

3) The authors do not explicitly define how stigma is measured-although it appears to be measured by the social distance variable. The operationalization of stigma must be explicit and justified.

ad 3.) Stigma was assessed via the social distance scale. We are aware that social distance does not capture all aspects of stigma. We clarified this in the measures section.

4) At one point, the authors talk about the positive effects of labeling and its effect on social distancing. They mention that pity/sympathy might be an effect of the label, thereby reducing the wish for social distancing. In the world of disability studies, there is nothing worse than pity. The rallying cry of disability advocates in the disability rights movement was "piss on pity". I cannot see how social contact, as a result of pity, would be a positive.

ad 4.) We agree, and don’t think (and don’t say) that the contact resulting out of pity or sympathy would be a positive one. Among other interpretations, we offered this one as a possible cause for the associations found in our study data. But we agree with the reviewer that pity cannot lead to a positive contact with a person. In the revised manuscript, we made sure that this sentiment was added in a clearer way..

5) Notion of requiring expert help as destigmatizing---is reflective of the medical model of disability in which disabled people are "sick" and need of a "cure". This in itself is stigmatizing as perceived by the disability rights community.

ad 5.) We agree with the reviewer, and made changes accordingly in the discussion section.

6) Not convinced that differences in attitudes by country is a result of Nazi past. UK and Germany have very distinct health care systems and welfare regimes. More importantly, the UK had a very visible, powerful disability rights movement--in fact British disabled advocates coined the social model of disability. They were leaders in the field. The movement essentially began in Britain and inspired US advocates to forge their own movement. This is not mentioned and likely plays a role in public attitudes. Failure to adequately address the disability rights movement--and its impact on attitudes-- is a major omission in this manuscript.

ad 6.) We added the disability rights movement to the introduction and discussion section.

Thank you again for the opportunity to review the manuscript. I think the study could be improved by greater understanding and inclusion of the disability studies literature and disability rights movement.

Reviewer #2: 

This is an interesting study that compared intellectual disability literacy and stigma among adults in the general population in the UK, Austria and Germany, and examined the effects of providing an intellectual disability label on these outcomes. Stigma is still an important societal issue. The paper is clearly written and methodologically sound.

 Thank you for this precious evaluation!

I have few comments / questions

Subject demographics:

How where subjects recruited exactly? Authors state that for the UK sample they used a stratified sample selected from a larger data set and that potential participants were approached either in person or via email circulation lists with a link to an anonymous web survey.

How was the original sample recruited and for what purpose? Is this a representative sample for larger society? Please provide more information.

We are sorry for the lack of clarity. We added more details to the methods section. In the procedures section we explained the sampling strategy in more detail. As we used a convenience sampling method in both localities, the samples cannot be regarded representative for a larger population. We also added this information in the procedure section, and included this aspect as a limitation in the discussion.

In addition, were other demographic variables that could influence the outcome measures such as ethnicity or religion assessed? Did participants had relatives with ID of were mental health professionals? Such factors could have influences participants ID literacy and stigma.

Thank you for this very thoughtful comment! Unfortunately, there are some demographic aspects, that were assessed only in the UK, like ethnicity. We did not assess if participants had a relative with ID or were health professionals. In the revised manuscript we discuss these aspects in the limitations section. 

However, we did assess religion in all countries, and now report this via a dichotomous variable (religious vs. non-religious). We included this variable in the description of the participants demographics, and used it as an additional predictor for social distance in the regression analysis. 

The authors stat that it was not possible to calculate a response rate, given that the number of persons who received an email invitation could not be tracked. This seems a bit odd, given that they stratified a sample based on a previous data set (at least for UK sample).

Again, we are sorry for the lack of clarity regarding sampling strategy. As mentioned earlier, we aimed to remedy this by including further details to the methods section. The UK data was randomly stratified from an existing database (participants were not contacted again, only the existing data was stratified). We did use the same initial sampling procedure in the UK, Austria and Germany: The link to the online survey was distributed and could be forwarded by any participant. Therefore, we have no records on how many persons received an invitation for the study. We could calculate the proportion of persons who completed the questionnaire after clicking on the invitation link. However, we don’t feel that this is a reliable number for the following reasons: a) people may click on a link just for information; b) some may come back later to complete the questionnaire – clicking the link again; and c) some people receiving the invitation will never klick on the link. Therefore, we deemed this number (the only number we could calculate for a response rate) as not suitable to resemble a valid response rate and decided not to use this information. We hope you share our assessment in this matter.

Did the authors also explore differences in outcome measures for the Austrian and German sample? Although generally comparable, there are likely to be more subtle cultural differences or differences in government polices possibly affecting the outcome measures.

We examined differences between the Austrian and German sample for all outcome measures. As there were no significant differences (alpha level = 5%), we combined them for all further analysis. This information can be found in the section describing the participants.

Can the authors provide the vignette and the causal items participants had to rate? Perhaps in a supplement. Did the vignette also include behavioral symptoms?

We include the English and German language vignettes as a supplement.

---

## [Decision Letter · Decision Letter 1]

16 Sep 2020

Intellectual disability literacy and its connection to stigma: A multinational comparison study in three European countries

PONE-D-20-07449R1

Dear Dr. Zeilinger,

We’re pleased to inform you that your manuscript has been judged scientifically suitable for publication and will be formally accepted for publication once it meets all outstanding technical requirements.

Kind regards,

Therese van Amelsvoort

Academic Editor

PLOS ONE

Additional Editor Comments (optional):

Reviewers' comments:

Reviewer's Responses to Questions

**Comments to the Author**

1. If the authors have adequately addressed your comments raised in a previous round of review and you feel that this manuscript is now acceptable for publication, you may indicate that here to bypass the “Comments to the Author” section, enter your conflict of interest statement in the “Confidential to Editor” section, and submit your "Accept" recommendation.

Reviewer #2: All comments have been addressed

2. Is the manuscript technically sound, and do the data support the conclusions?

Reviewer #2: Yes

3. Has the statistical analysis been performed appropriately and rigorously? 

Reviewer #2: Yes

4. Have the authors made all data underlying the findings in their manuscript fully available?

Reviewer #2: Yes

5. Is the manuscript presented in an intelligible fashion and written in standard English?

Reviewer #2: Yes

6. Review Comments to the Author

Reviewer #2: (No Response)

7. PLOS authors have the option to publish the peer review history of their article (what does this mean?). If published, this will include your full peer review and any attached files.

Reviewer #2: No

---

## [Editor Report · Acceptance letter]

6 Oct 2020

PONE-D-20-07449R1 

Intellectual disability literacy and its connection to stigma: A multinational comparison study in three European countries 

Dear Dr. Zeilinger:

I'm pleased to inform you that your manuscript has been deemed suitable for publication in PLOS ONE. Congratulations! Your manuscript is now with our production department. 

Kind regards, 

on behalf of

Prof. Therese van Amelsvoort 

Academic Editor

PLOS ONE